# The efficacy of topical, oral and surgical interventions for the treatment of tungiasis: A systematic review of the literature

**Ana Carolina Tardin Martins**[1], **Amanda Ramos de Brito**[2], **Patrícia Shu Kurizky**[1], **Rodrigo Gurgel Gonçalves**[3], **Yago Ranniere Teixeira Santana**[4], **Fabiola Christian Almeida de Carvalho**[2], **Ciro Martins Gomes**[1,3]*

1 Programa de Pós-Graduação em Ciências Médicas, Faculdade de Medicina, Universidade de Brasília, Brasília, Brazil, 2 Programa de Pós-Graduação em Ciências da Saúde, Universidade Federal de Roraima, Boa Vista, Brazil, 3 Programa de Pós-Graduação em Medicina Tropical, Núcleo de Medicina Tropical, Universidade de Brasília, Brasília, Brazil, 4 Secretaria Especial de Saúde Indígena (Sesai), Ministério da Saúde do Brasil, Brasília, Brazil

* cirogomes@unb.br

**Data Availability Statement:** All relevant data are within the manuscript and its Supporting information files.

## Abstract

### Background

Tungiasis is a neglected disease caused by *Tunga penetrans* that can be complicated by secondary infections and local tissue destruction. Adequate treatment is important, especially in vulnerable populations; potential treatment options proposed range from surgical extraction to the use of oral and topical medications. We aimed to perform a systematic review to assess the efficacy of topical, oral and surgical interventions for the treatment of tungiasis.

### Methodology/Principal findings

The present review is registered in PROSPERO (CRD42021234741). On September 1, 2020, we searched PubMed, EMBASE, Scopus, Web of Science, Science Direct, Scielo and LILACS BVS. We included clinical trials and longitudinal observational studies that evaluated any topical, systemic or mechanical treatment for tungiasis. We used the Revised Cochrane Risk of Bias (RoB) Tool for Randomized Trials for clinical trial analysis. Qualitative and quantitative descriptive syntheses were performed. Our search strategy resulted in 3376 references. Subsequently, 2568 titles/abstracts and 114 full texts were screened. We finally included 19 articles; 9 were classified as clinical trials. Two and 3 articles presented low and some RoB, respectively, according to the tool. Only two articles tested the efficacy of oral medications (niridazole, ivermectin), with discouraging results. Six clinical trials evaluated topical products for the treatment of tungiasis; 2 evaluated dimeticone-based compounds and reported positive results in lesion reduction and cure. None reported significant adverse reactions. Surgical extraction was evaluated only in observational studies.

**Funding:** CMG - This work was supported by Conselho Nacional de Desenvolvimento Científico e Tecnológico e Ministério da Saúde, MS-SCTIE-Decit No. 22/2019, Brazil. The funders had no role in study design, data collection and analysis, decision to publish, or preparation of the manuscript.

**Competing interests:** The authors have declared that no competing interests exist.

## Conclusions/Significance

We conclude that, although surgical extraction is the most commonly used treatment, there is sufficient evidence supporting the use of occlusive agents, especially manufactured dimeticone-based products.

## Author summary

Tungiasis is a disease caused by *Tunga penetrans* that affects regions with low socioeconomic status and a lack of proper sanitation. The disease usually has a self-limiting course or can be cured by simple extraction, but complications are not uncommon. In vulnerable populations, such as indigenous communities, children and people with disseminated tungiasis, the development of new treatment strategies is essential for the prevention of undesirable secondary outcomes. We performed a comprehensive systematic review of the literature by searching the most important scientific databases: PubMed, EMBASE, Scopus, Web of Science, Science Direct, Scielo and LILACS BVS. We aimed to assess the efficacy of topical, oral and surgical interventions for the treatment of tungiasis. We included 19 articles, 9 of which were classified as clinical trials. Six clinical trials evaluated topical products for the treatment of tungiasis; 2 evaluated dimeticone-based compounds and reported positive results in lesion reduction and cure. None reported significant adverse reactions. We concluded that, although mechanical extraction is the most commonly used treatment, there is sufficient evidence supporting the use of occlusive agents, especially manufactured dimeticone-based products.

## Introduction

Tungiasis is a neglected disease caused by *Tunga penetrans* endemic to areas with a lack of sanitation and sandy soils, with an important prevalence in Latin America and sub-Saharan Africa. The disease occurs more frequently in travellers, riverside communities, slums in large urban centres, indigenous communities and rural communities.[1].

Typically, *T. penetrans* affects the periungual region of the toes and heels but can affect other parts of the body, such as the hands, elbows, buttocks, and legs. According to the Fortaleza classification,[1,2] the disease can be described in 5 stages: 1) penetration, in which *T. penetrans* partially penetrates the skin; the main symptom is itching; 2) the beginning of parasite hypertrophy, in which a 0.5- to 2-mm central brown-coloured spot surrounded by an erythematous area appears; 3) maximal hypertrophy, in which a white circular zone with a diameter of three to ten mm with a small central black dot appears; the most common symptoms in this phase are erythema, oedema, tenderness, heat, pain, intense itching and flaking of the corneal layer around the lesion; 4) initial involution, in which involution of the lesion occurs, and the hypertrophic zone decreases at 3 weeks after penetration, with improvement in the condition after 5 weeks on average; and 5) expulsion, in which the parasite is expelled from the body; clinical residues can be observed from week 6 until several months after penetration.[2].

Ectoparasitosis is usually self-limiting, but secondary complications are not uncommon. These complications can include fissures, leading to pain and difficulty walking, as well as finger deformities and nail loss. In addition, secondary infections can develop as a complication. Tungiasis lesions usually occur in groups and have a growth pattern that resembles tumour

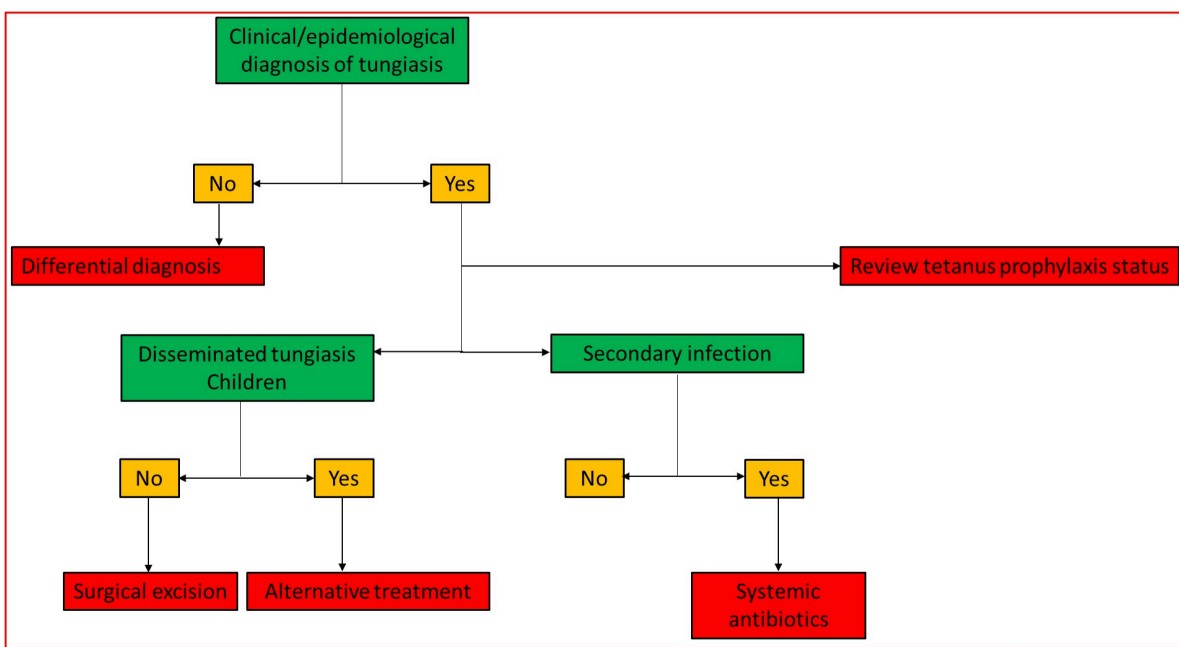

**Fig 1. Current clinical pathway proposed for the treatment of tungiasis.**

formation. Injuries caused by *T. penetrans* can serve as entry points for other diseases[3] such as tetanus, which can cause death.[4] In addition to biological problems, social problems are common in affected communities.[4–6] Special populations, such as indigenous communities and immunosuppressed patients, are also at a higher risk of complications.[5,7].

Several treatments have been proposed in the literature, from surgical excision of the flea to the use of anthelmintics and topical medications. Surgical removal requires the use of adequate and sterile instruments. Such resources may be scarce in communities with low socioeconomic status. In addition, extraction can be extremely painful, especially for multiple lesions.[6,8] Topical and oral medications, such as dimeticones, ivermectin and thiabendazole, have been evaluated as possible alternatives to surgical extraction.[8,9] It is important to develop a clinical pathway for proper understanding of disease management (Fig 1). Additionally, the identification of significant gaps in the existing literature is important for future research, especially for neglected diseases.

Since there has been a clear scarcity of investments and research in the field of ectoparasitosis, we aimed to perform a comprehensive systematic review of the literature to assess the efficacy of topical, oral and surgical interventions for the treatment of tungiasis. We also aimed to analyse the current methodological approaches in the most relevant studies and to discuss clinical characteristics related to tungiasis and its impact in specific populations, such as children and indigenous communities.

## Methods

### Search strategy and article selection

We strictly followed a predesigned review protocol registered in PROSPERO (CRD42021234741) and the PRISMA 2020 statement: an updated guideline for reporting systematic reviews (S1 PRISMA Checklist)[10]. On September 1, 2020, we comprehensively searched for articles in PubMed, EMBASE, Scopus, Web of Science, Science Direct, Scielo and

**Table 1. Databases, websites and search terms used in the review.**

| Databases | Search Terms |
|---|---|
| PubMed (https://pubmed.ncbi.nlm.nih.gov) | (human*) AND ((tunga*) OR (tungiasis)) |
| Embase (https://www.embase.com) | (tungiasis OR tunga*) AND humans |
| Scopus (https://www.scopus.com) | ((ALL (tunga) OR ALL (tungiasis))) AND (ALL (human*)) |
| Science Direct (https://www.sciencedirect.com) | (tunga OR tungiasis) AND human |
| Scielo (https://scielo.org) | ((tunga*) OR (tungiasis)) AND (human*) |
| LILACS BVS (https://lilacs.bvsalud.org/) | tw:(((tunga*) OR (tungiasis)) AND (human*)) AND (db: ("LILACS")) |
| Web of Science (https://app.webofknowledge.com/author/search) | (tunga* OR tungiasis) AND human* |

LILACS BVS. Complete details of the search strategy are described in Table 1. Grey literature, including meeting abstracts, was searched using the aforementioned databases, and articles in thesis databases were also searched (S1 File). Additional references were identified using a backward snowballing method.[11] No restrictions regarding language or publication date were applied.

We included any type of clinical trial or longitudinal observational study that assessed patients diagnosed with tungiasis and evaluated any topical treatment, systemic treatment or mechanical extraction methods for parasite eradication. We excluded case series and case reports because these types of studies failed to provide analytical conclusions related to the treatment options. We also excluded trials that evaluated only environmental strategies for the control of tungiasis. All of the selected titles and abstracts were exported to EPPI-Reviewer 4 Version 4.6.4.0 (EPPI Centre, London, UK), and duplicates were removed. Two independent reviewers (CG & AM) screened the titles and abstracts and subsequently screened the selected full texts. Disagreements regarding title and abstract evaluation and full-text evaluation were resolved by a third independent reviewer (PK).

## Critical analysis instruments

We evaluated the possible risk of bias (RoB) in clinical trials using the Revised Cochrane Risk of Bias Tool for Randomized Trials. For observational studies, we used the Critical Appraisal Tool from the Joanna Briggs Institute. Both analyses were performed by two independent reviewers (CG & AM), and disagreements were resolved by consensus.

## Data extraction, classification and analysis

Data extraction was performed after the creation of an extraction form based on the clinical experience of the researchers (S1 Table). The form collected important information about the publication year, the geographical area where the study was conducted, the number of patients included, the inclusion of indigenous communities or children, the treatment of disseminated tungiasis, the presence of complications related to tungiasis, the type of treatment and the treatment outcomes. Data extraction was performed by two independent reviewers (CG & AM), and disagreements were resolved by consensus.

In the first step, a qualitative analysis of the included articles was performed without any limitations regarding the minimum number of articles. The information collected using our extraction form was evaluated, and the impacts of those characteristics were considered based on expert opinion. The definition of disseminated tungiasis followed the discretion of each

selected article. Moreover, the possible impacts of the studied variables in future trials were also considered.

Considering that most articles presented great variability in outcome measures, the quantitative synthesis was based on frequencies and on individual relative risks. No additional quantitative synthesis method was performed due to the evident heterogeneity in article methods, intervention reporting, outcome measurements and quality analysis.

## Results

Our search strategy resulted in 3376 references. We excluded 808 duplicates and evaluated 2568 titles and abstracts. Subsequently, we screened 114 full texts (11 were not retrieved) and finally included 19 articles. Nine articles reported the results of clinical trials for tungiasis treatments,[8,9,12–18] and ten reported the results of observational studies (Fig 2).[3,5,19–26].

### Population overview

The articles included 2796 treated patients from 1982 to 2020 (S1 Table). In one article, the authors used individual and environmental interventions, making it difficult to calculate the number of people who benefited.[16] Studies were performed in Argentina (number of articles (n) = 1),[19] Brazil (n = 6),[3,8,9,13,23,26] Colombia (n = 1),[5] Ethiopia (n = 1),[24] Haiti (n = 1),[16] Kenya (n = 2),[14,18] Madagascar (n = 2),[20,25] Nigeria (n = 2),[12,15] Tanzania (n = 1),[22] Trinidad and Tobago (n = 1)[21] and Uganda (n = 1).[17] Seventeen studies evaluated treatment in children, and only one evaluated an indigenous community. Disseminated tungiasis was reported in 9 studies, and 9 studies reported the occurrence of secondary infections, including sepsis. Deformities and amputations were also described.

### Critical analysis results

The results of the RoB analysis of clinical trials and critical appraisal of observational studies are shown in Fig 3 and S2 Table, respectively. According to our analysis, 2 articles had a low

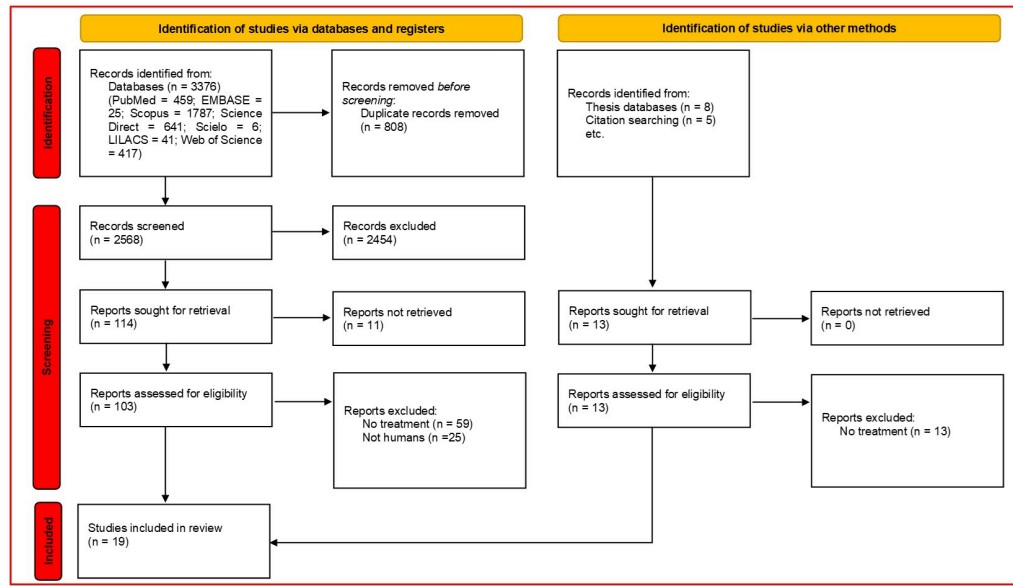

**Fig 2. PRISMA flow diagram of this review.**

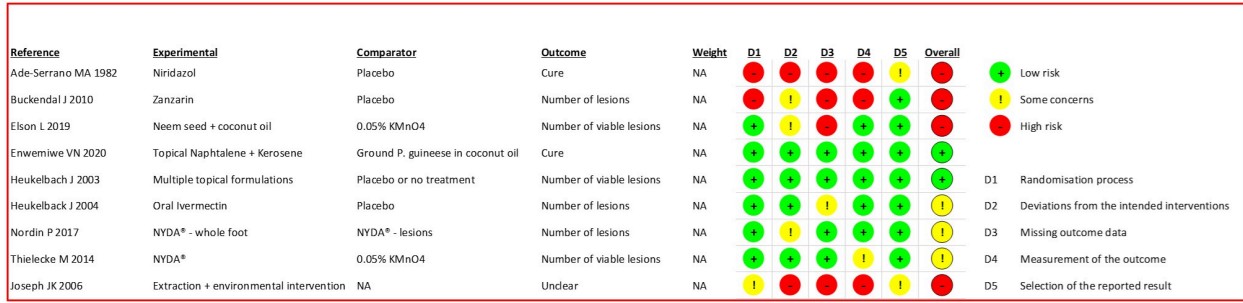

| Reference | Experimental | Comparator | Outcome | Weight | D1 | D2 | D3 | D4 | D5 | Overall | | |
|---|---|---|---|---|---|---|---|---|---|---|---|---|
| Ade-Serrano MA 1982 | Niridazol | Placebo | Cure | NA | ● | ● | ● | ● | ! | ● | + | Low risk |
| Buckendal J 2010 | Zanzarin | Placebo | Number of lesions | NA | ● | ! | ! | ● | + | ● | ! | Some concerns |
| Elson L 2019 | Neem seed + coconut oil | 0.05% KMnO4 | Number of viable lesions | NA | + | ! | ● | + | + | ● | ● | High risk |
| Enwemiwe VN 2020 | Topical Naphtalene + Kerosene | Ground P. guineese in coconut oil | Cure | NA | + | + | + | + | + | + | | |
| Heukelbach J 2003 | Multiple topical formulations | Placebo or no treatment | Number of viable lesions | NA | + | + | + | + | + | + | D1 | Randomisation process |
| Heukelbach J 2004 | Oral Ivermectin | Placebo | Number of lesions | NA | + | + | ! | + | + | ! | D2 | Deviations from the intended interventions |
| Nordin P 2017 | NYDA® - whole foot | NYDA® - lesions | Number of lesions | NA | + | ! | + | + | + | ! | D3 | Missing outcome data |
| Thielecke M 2014 | NYDA® | 0.05% KMnO4 | Number of viable lesions | NA | + | + | + | ! | + | ! | D4 | Measurement of the outcome |
| Joseph JK 2006 | Extraction + environmental intervention | NA | Unclear | NA | ! | ● | ● | ● | ! | ● | D5 | Selection of the reported result |

**Fig 3. Graphic presentation of the quality evaluation of included clinical trials using the Revised Cochrane Risk of Bias Tool for Randomized Trials.**

overall RoB according to the Cochrane Risk of Bias Tool for Randomized Trials.[8,15] Three articles had some RoB: 1 because of a small deviation from the intended intervention,[9] 1 because of missing outcome data[17] and 1 because of the outcome measurement.[18] These five studies were considered to have satisfactory quality for each theme because the RoB tool used was considered very rigorous and aimed to cover all aspects of clinical trials. In fact, most concerns were related to the proper natural history of tungiasis. Heukelbach et al. (2004) transferred patients to a mountain resort for 8 days to eliminate bias from active environmental sources of infection, resulting in some follow-up losses.[9] The other 2 articles generated minimal concerns using interventions that were clearly different from each other, rendering blinding of the evaluators and patients difficult; however, we considered that there was no influence on the outcome measurement (Fig 3).[17,18] We also observed that the evaluation of outcomes can be difficult in tungiasis cases due to difficulties in evaluating individuals or the number/viability of lesions.

In the critical analysis of observational studies, although interventions were more adequately evaluated in clinical trials, 2 articles positively scored "YES" in all domains of the Critical Appraisal Tools from the Joanna Briggs Institute.[21,26] Recognition of the disease was not a problem in either clinical trials or observational studies because the disease courses had characteristic epidemiological features and clinical presentations. The sampling methods were unclear in 6 articles, rendering any assumption about disease prevalence unfeasible (S2 Table).

## Treatment overview

All of the observational studies described extraction methods as the most frequent type of treatment used in populations. Unfortunately, information about treatment success and adverse reactions was scarce in observational studies (Table 2). Details about the interventions evaluated in clinical trials are shown in Table 3. Only two articles tested the efficacy of oral medications in the treatment of tungiasis. Ade-Serrano et al. (1982) tested the utility and safety of niridazole against a placebo.[12] In addition to gastrointestinal effects, a limitation related to reinfection in an uncontrolled environment was noted by the authors. Heukelbach et al. (2004) tested oral ivermectin against a placebo, with discouraging results.[9].

Six additional studies evaluated topical applications of products with occlusive properties or substances with direct parasiticidal effects. None of the 6 studies reported significant adverse reactions, and most demonstrated variable beneficial effects of the tested interventions on tungiasis (Table 3). Interestingly, 2 studies evaluated the topical application of the manufactured product NYDA (Pohl-Boskamp GmbH & Co. KG, Hohenlockstedt, Germany), with no reports of relevant side effects and with interesting properties related to

**Table 2. Study characteristics of the included observational studies.**

| Reference | Country | n | Indigenous communities | Children | Disseminated tungiasis | Infection | Treatment |
|---|---|---|---|---|---|---|---|
| Chadee DD 1998 | Trinidad and Tobago | 268 | No | Yes | No | Yes | Extraction |
| Oscherov B 2008 | Argentina | 124 | No | Yes | No | No | Extraction |
| Belaz S 2015 | Madagascar | 16 | No | Yes | No | Yes | Extraction |
| Dassoni F 2014 | Tanzania | 62 | No | Yes | Yes | Yes | Extraction |
| de Carvalho RW 2003 | Brazil | 132 | No | Yes | Yes | Yes | Extraction |
| Feldmeier H 2004 | Brazil | 86 | No | Yes | Yes | Yes | Extraction |
| Girma M 2018 | Ethiopia | 366 | No | Yes | Yes | Yes | Extraction and natural products |
| Miller H 2010 | Colombia | 942 | Yes | Yes | Yes | Yes | Extraction |
| Schuster A 2017 | Madagascar | 36 | No | Yes | No | No | Extraction |
| Winter B 2009 | Brazil | Unclear | No | No | No | No | Extraction |

efficacy.[17,18] An additional study tested the effect of mechanical extraction in addition to environmental interventions on the control of tungiasis, calling attention to this important technique for the control of the disease. Unfortunately, the detailed data of each individual were not reported.[16].

**Table 3. Details of the interventions evaluated in the 8 included clinical trials.** Different comparisons within the same study were considered different levels of analysis and are included in different lines.

| Author | Experimental | Comparator | Ev Experimental | n Experimental | Ev Comparator | n Comparator |
|---|---|---|---|---|---|---|
| Ade-Serrano MA 1982 | 2x niridazole 30 mg/kg (1 week interval) | Placebo (ascorbic acid) | 49 | 49 | 0 | 28 |
| Buckendahl J 2010 | Zanzarin 2x/day/4 weeks + 2x/day in alternate weeks/5 months | No intervention | Nl | 43 | Nl | 41 |
| Buckendahl J 2010 | Zanzarin 2x/day/4 weeks + 2x/day/week/1 week-month for 5 months | No Intervention | Nl | 33 | Nl | 41 |
| Elson L 2019 | Cold-pressed 20% virgin neem seed oil + 80% virgin coconut oil on days 1 and 3 | 0.05% KMnO4 solution for 15 minutes | Vl | 48 | Vl | 45 |
| Enwemiwe VN 2020 | Naphthalene and kerosene mixture 2x/day/ 2 weeks | Ground *Piper guineese* in coconut oil topical ointment 2x/day/2 weeks | 40 | 40 | 40 | 40 |
| Heukelbach J 2003 | Ivermectin lotion (0.8%, w/v) | Placebo lotion or no treatment | Vl | 33 | Vl | 52 |
| Heukelbach J 2003 | Metrifonate (trichlorfon) lotion 0.2%, (w/v) | Placebo lotion or no treatment | Vl | 24 | Vl | 52 |
| Heukelbach J 2003 | Thiabendazole lotion (5%, w/v) | Placebo lotion or no treatment | Vl | 33 | Vl | 52 |
| Heukelbach J 2003 | Thiabendazole ointment (5%, w/v) | Placebo lotion or no treatment | Vl | 27 | Vl | 52 |
| Heukelbach J 2004 | 2x 300 µg/kg body weight ivermectin | Placebo | Nl | 27 | Nl | 27 |
| Joseph JK 2006 | Mechanical extraction + environmental intervention | - - | - - | - - | - - | - - |
| Nordpin P 2017 | NYDA to the plantar side of one foot | (NYDA) 3 times within 10 minutes on each lesion | Nl | | Nl | |
| Thielecke M 2014 | NYDA | 0.05% KMnO4 solution | Vl | | Vl | |

Ev = number of events related to the measured outcome; n = number of patients; Nl = measurement of the reduction in the number of lesions; Vl = measurement of the reduction in the number of viable lesions.

## Discussion

Tungiasis is a neglected disease related to low socioeconomic status and unfavourable living conditions. The geographical distribution of the included studies is evidence that the disease occurs in communities with severe limitations on sanitation and economic development. Although the final number of studies (n = 19) included in the review was not extremely small, our search strategy revealed that tungiasis has been neglected to a greater extent than diseases such as leishmaniasis and leprosy. Even with the use of a comprehensive search strategy, comprehensive terms to describe the disease, term expansion and no limiting operators, only 2568 titles and abstracts were identified in the initial screening. These results indicate sub-notification of the disease and a lack of proper investment in its control.

The geographical distribution of the included studies was limited to South American and African countries. These regions are tropical with widely known economic problems. The association between a climate favourable for the proliferation of *T. penetrans* and inadequate sanitation is the main risk factor for the occurrence of the disease. In fact, the importance of the environment was clearly noted. Most of the clinical trials that had some concern and a high risk of bias in the outcome measurement did not use adequate strategies to control *T. penetrans* in the environment. We believe that the treatment of tungiasis could be clearly ineffective if the environmental infection source is not addressed. This fact makes the conduction of clinical trials for tungiasis treatment a specific and difficult task. Heukelbach et al. (2004) isolated trial participants in an area free of the infection source.[9] The authors recognized that this fact could be a limiting factor in the efficacy evaluations of the tested interventions. However, some patients had to return to their homes, representing losses to follow-up.

No trials directly compared oral versus topical drugs or any drug versus mechanical extraction. However, we observed some limitations related to oral niridazole (gastrointestinal effects) and oral ivermectin (reported lack of efficacy). Oral drugs with anti-parasitic effects are always considered for the treatment of cutaneous ectoparasitoses because of their easy administration. Single-dose oral drug administration, in addition to environmental control, seems to be an interesting strategy for tungiasis control, especially in remote communities. However, no studies reporting the success of this combined intervention were found.

Topical drugs were the most successful treatments in our systematic review of the literature. Although the exact mechanism of action of such drugs must still be determined, most of the included drugs had an occlusive effect on parasites. In most cases, we could not determine whether the positive results of trials evaluating topical drugs were related to the active principle or to the occlusive property inherent to the treatment used. Despite this limitation, the rates of relevant adverse reactions were exceptionally low for all topical drugs, and adverse reactions are seemingly not a relevant problem in the use of the tested topical drugs. However, to evaluate the real efficacy of these drugs, parallel trials comparing topical drugs to their active ingredients alone must be performed.

Dimeticones, a family of silicon oils, are the most studied compounds. A recent narrative review of the literature described the mechanism of action of dimeticones against ectoparasites in detail.[27] Interestingly, 2 well-conducted clinical trials tested the manufactured drug NYDA, a 92% dual-formula dimeticone-based drug approved for infants and pregnant/breast-feeding women in Germany, with encouraging results related to the efficacy and the apparent absence of relevant adverse effects after recommended use. According to previous proof-of-principle data, the mechanism of action of dimeticone is purely physical.[14] The substance occludes the tracheae of *T. penetrans*, resulting in parasite death.[18] The efficacy of this mechanism depends on the infection evolutionary phase of the parasite in human skin. Parasites in early stages of development, which are in an intensive metabolic state, are more susceptible to

occlusive treatments, while fully developed parasites may have a less pronounced response. [18] In clinical studies, it was applied to the whole affected foot and was also applied directly to each lesion; lesion-specific application showed better efficacy and represents a cost-effective strategy.[17] An effective manufactured drug tested in two clinical trials is an important treatment option for patients with tungiasis. In addition to having shown adequate results in clinical trials, production quality and consistency can be monitored. This goal cannot adequately be achieved with natural products that are not registered with regulatory organizations.

Although a relatively small number of scientific studies is always a limitation for systematic reviews targeting the treatment of neglected diseases, we selected 5 studies that did not present a high RoB. Limitations related to high heterogeneity among comparisons and outcomes preclude any detailed quantitative synthesis and could jeopardize the evaluation of reproducibility of the results. However, we believe that there is sufficient evidence to recommend topical agents with occlusive and physical properties for the treatment of tungiasis, especially dimeticone-based compounds. The finding of 2 well-designed trials testing the manufactured drug NYDA is reassuring, especially for complex cases, such as disseminated tungiasis and tungiasis in children for whom mechanical extraction is difficult.

We conclude that, although tungiasis is a neglected disease, and mechanical extraction is the most commonly used type of treatment, there is sufficient evidence supporting the use of occlusive agents, especially manufactured dimeticone-based products. Tungiasis-endemic countries must provide adequate approval and regulation of these products to prevent complications related to improper treatment. We conclude that it is also paramount for the control of tungiasis that future investments and studies be accompanied by environmental interventions to eliminate *T. penetrans* as a source of infection.

## Supporting information

**S1 PRISMA Checklist.**
(DOCX)

**S1 File. Web-based links to all of the thesis databases searched.**
(DOCX)

**S1 Table. Table containing the complete data extracted in the review process.**
(XLSX)

**S2 Table. Critical appraisal tool from the Joanna Briggs Institute.**
(XLSX)

## Acknowledgments

We thank all of the employees of the Pan American Health Organization (PAHO)–Brasília, Brazil, and of Secretaria Especial de Saúde Indígena (Sesai), Ministério da Saúde do Brasil, Brazil, for their unconditional support in this effort.

## Author Contributions

**Conceptualization:** Yago Ranniere Teixeira Santana, Fabiola Christian Almeida de Carvalho, Ciro Martins Gomes.

**Data curation:** Ana Carolina Tardin Martins, Yago Ranniere Teixeira Santana, Fabiola Christian Almeida de Carvalho, Ciro Martins Gomes.

**Formal analysis:** Ana Carolina Tardin Martins, Ciro Martins Gomes.

**Funding acquisition:** Ciro Martins Gomes.

**Investigation:** Amanda Ramos de Brito, Patrícia Shu Kurizky, Ciro Martins Gomes.

**Methodology:** Ana Carolina Tardin Martins, Amanda Ramos de Brito, Patrícia Shu Kurizky, Ciro Martins Gomes.

**Project administration:** Fabiola Christian Almeida de Carvalho, Ciro Martins Gomes.

**Resources:** Patrícia Shu Kurizky, Ciro Martins Gomes.

**Software:** Ciro Martins Gomes.

**Supervision:** Ciro Martins Gomes.

**Validation:** Patrícia Shu Kurizky, Rodrigo Gurgel Gonçalves, Ciro Martins Gomes.

**Visualization:** Rodrigo Gurgel Gonçalves, Yago Ranniere Teixeira Santana, Ciro Martins Gomes.

**Writing – original draft:** Ana Carolina Tardin Martins, Ciro Martins Gomes.

**Writing – review & editing:** Rodrigo Gurgel Gonçalves, Fabiola Christian Almeida de Carvalho, Ciro Martins Gomes.

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
