## [Decision Letter · Decision Letter 0]

30 Jul 2021

Dear Dr. Gomes,

Thank you very much for submitting your manuscript "The efficacy of topical, oral and surgical interventions for the treatment of tungiasis: a systematic review of the literature." for consideration at PLOS Neglected Tropical Diseases. As with all papers reviewed by the journal, your manuscript was reviewed by members of the editorial board and by several independent reviewers. The reviewers appreciated the attention to an important topic. Based on the reviews, we are likely to accept this manuscript for publication, providing that you modify the manuscript according to the review recommendations. 

Sincerely,

Joseph M. Vinetz

Deputy Editor

Joseph Vinetz

Deputy Editor

Reviewer's Responses to Questions

**Key Review Criteria Required for Acceptance?**

**Methods**

-Are the objectives of the study clearly articulated with a clear testable hypothesis stated?

-Is the study design appropriate to address the stated objectives?

-Is the population clearly described and appropriate for the hypothesis being tested?

-Is the sample size sufficient to ensure adequate power to address the hypothesis being tested?

-Were correct statistical analysis used to support conclusions?

-Are there concerns about ethical or regulatory requirements being met?

Reviewer #1: (No Response)

Reviewer #2: The authors aimed to to assess the efficacy of currently available topical, oral, and surgical interventions for the treatment of tungiasis, which is clearly delineated by the hypothesis. The study design is appropriate, with enough articles included for a PRISMA-compliant systematic review. Furthermore, the authors followed all applicable regulatory guidelines during this study.

Reviewer #3: The purpose of this paper was to collect information and find the most efficacious treatment for Tungiasis, and this paper accomplishes that very well.

**Results**

-Does the analysis presented match the analysis plan?

-Are the results clearly and completely presented?

-Are the figures (Tables, Images) of sufficient quality for clarity?

Reviewer #1: (No Response)

Reviewer #2: The presented results appear to be more about methods than key findings and should be updated. Furthermore, the information in the tables is detailed and easy to understand. The figures are adequate and of sufficient quality in accordance with the journal's guidelines.

Reviewer #3: The results and analysis were clear and the treatment overview was extremely helpful.

**Conclusions**

-Are the conclusions supported by the data presented?

-Are the limitations of analysis clearly described?

-Do the authors discuss how these data can be helpful to advance our understanding of the topic under study?

-Is public health relevance addressed?

Reviewer #1: (No Response)

Reviewer #2: Significant findings of this study support conclusion. This study's limitations are appropriately described. More robust discussion about the study's potential prospects for advancing public health relevance is required.

Reviewer #3: The conclusions are directly supported by the data. Limitations were explained and presented accordingly. The support of Dimeticone as treatment is well founded but in this paper there is a lack of explanation. The occlusive action is mentioned but not explained. I would include a section explaining the method of action of this treatment. I would also suggest that the authors include a image of the the different types of treatments after application, and if possible a graphic depicting what a individual Tunga penetrans looks like and why the occlusive nature of this treatment works.

**Editorial and Data Presentation Modifications?**

Reviewer #1: (No Response)

Reviewer #2: (No Response)

Reviewer #3: The paper does need some minor grammatical revisions for more concise wording and make the paper easier to read.

**Summary and General Comments**

Reviewer #1: Manuscript PNTD-D-21-00651 reviewed the progress of treatment of the neglected disease tungiasis mostly occurred in South America and Africa. The clinical trials included in this review article supported the application of topical drugs such as dimeticones as the encouraging yet cost-effective therapy for the treatment of this disease. The paper is well organized and written. The conclusions drawn from this review paper are of great values for the clinical trials as well as the regulatory purposes in the affected countries. I recommend to accept this manuscript without further changes.

Reviewer #2: This is a well-executed important study with apparent strength. The review is registered prospectively, and the authors investigate the efficacy of various interventions for the treatment of tungiasis. To avoid publication and language bias, the review employs the Cochrane Risk of Bias Tool and the Critical Appraisal Tool. This article's rationale and novelty statement should be elaborated at the appropriate place. Here are the key issues with this article: What about the studies that were excluded during the qualitative analysis of the articles (case series, case reports, and trials)? How did they fail to contribute to the study, and how did their exclusion affect the study's outcome? The materials and methods must explicitly state how the studies' risk of bias and assessment of quality studies was judged. Please ensure that the search and inclusion of studies is as up to date as possible.

Reviewer #3: Overall this paper adequately accomplishes its goal. I do want to reiterate the need for a explanation of the method of action of this treatment. Understanding how this treatment works on a physical level when compared to other treatments is of paramount importance.

PLOS authors have the option to publish the peer review history of their article (what does this mean?). If published, this will include your full peer review and any attached files.

Reviewer #1: No

Reviewer #2: Yes: Amit Kumar Srivastava

Reviewer #3: Yes: Jonathan A. Niezgoda

Figure Files:

Data Requirements:

Reproducibility:

References

---

## [Editor Report · Decision Letter 1]

10 Aug 2021

Dear Dr. Gomes,

We are pleased to inform you that your manuscript 'The efficacy of topical, oral and surgical interventions for the treatment of tungiasis: A systematic review of the literature' has been provisionally accepted for publication in PLOS Neglected Tropical Diseases.

Best regards,

Joseph M. Vinetz

Deputy Editor

Joseph Vinetz

Deputy Editor

---

## [Editor Report · Acceptance letter]

17 Aug 2021

Dear Dr. Gomes,

We are delighted to inform you that your manuscript, "The efficacy of topical, oral and surgical interventions for the treatment of tungiasis: A systematic review of the literature," has been formally accepted for publication in PLOS Neglected Tropical Diseases.

Best regards,

Shaden Kamhawi

co-Editor-in-Chief

Paul Brindley

co-Editor-in-Chief
